# Cryo-EM structure of the SAGA and NuA4 coactivator subunit Tra1 at 3.7 angstrom resolution

Luis Miguel Díaz-Santín[1], Natasha Lukoyanova[2], Emir Aciyan[2], Alan CM Cheung[1,2]*

[1]Department of Structural and Molecular Biology, Institute of Structural and Molecular Biology, University College London, London, United Kingdom; [2]Institute of Structural and Molecular Biology, Biological Sciences, Birkbeck College, London, United Kingdom

**Abstract** Coactivator complexes SAGA and NuA4 stimulate transcription by post-translationally modifying chromatin. Both complexes contain the Tra1 subunit, a highly conserved 3744-residue protein from the Phosphoinositide 3-Kinase-related kinase (PIKK) family and a direct target for multiple sequence-specific activators. We present the Cryo-EM structure of *Saccharomyces cerevsisae* Tra1 to 3.7 Å resolution, revealing an extensive network of alpha-helical solenoids organized into a diamond ring conformation and is strikingly reminiscent of DNA-PKcs, suggesting a direct role for Tra1 in DNA repair. The structure was fitted into an existing SAGA EM reconstruction and reveals limited contact surfaces to Tra1, hence it does not act as a molecular scaffold within SAGA. Mutations that affect activator targeting are distributed across the Tra1 structure, but also cluster within the N-terminal Finger region, indicating the presence of an activator interaction site. The structure of Tra1 is a key milestone in deciphering the mechanism of multiple coactivator complexes.

DOI: https://doi.org/10.7554/eLife.28384.001

*For correspondence:
alan.cheung@ucl.ac.uk

**Competing interests:** The authors declare that no competing interests exist.

## Introduction

Cells execute precise programmes of transcription in response to environmental or developmental signals. These programmes are regulated by activator proteins which bind to specific DNA sequences and recruit coactivators that activate transcription by stimulating assembly of the basal transcriptional machinery and/or catalysing chromatin modifications at target genes (*Hahn and Young, 2011*; *Weake and Workman, 2010*). Coactivators often interact with multiple activators and are also targeted by signaling pathways, making them integrative hubs that interpret multiple inputs to modulate transcription (*Malik and Roeder, 2010*; *Rosenfeld et al., 2006*).

The yeast SAGA (Spt-Ada-Gcn5-Acetyltransferase) (*Grant et al., 1997*) and NuA4 (Nucleosome-Acetyltransferase-of-histone-H4) (*Allard et al., 1999*) coactivators are conserved in all eukaryotes but are evolutionarily and mechanistically unrelated to each other; SAGA is a 19–20 subunit, 1.8 MDa complex that stimulates preinitiation complex formation by interaction with TBP (*Dudley et al., 1999*), and contains H3 histone acetyltransferase (HAT) and H2B deubiquitinase enzymatic activities (*Daniel et al., 2004*; *Henry et al., 2003*), whereas NuA4 is a 13-subunit, 1.3 MDa complex that acetylates H4 and H2A (*Allard et al., 1999*; *Grant et al., 1997*). SAGA and NuA4 also have functions outside of transcription, with diverse but important roles in mRNA export (*Rodríguez-Navarro et al., 2004*), DNA repair (*Bird et al., 2002*; *Downs et al., 2004*) and telomere maintenance (*Atanassov et al., 2009*). Despite their differences, both complexes have integrated the large Tra1 (Transcription-Associated protein 1) subunit (*Allard et al., 1999*; *Grant et al., 1998*;

**eLife digest** Inside our cells, histone proteins package and condense DNA so that it can fit into the cell nucleus. However, this also switches off the genes, since the machines that read and interpret them can no longer access the underlying DNA. Turning genes on requires specific enzymes that chemically modify the histone proteins to regain access to the DNA. This must be carefully controlled, otherwise the 'wrong' genes can be activated, causing undesired effects and endangering the cell.

Histone modifying enzymes often reside in large protein complexes. Two well-known examples are the SAGA and NuA4 complexes. Both have different roles during gene activation, but share a protein called Tra1. This protein enables SAGA and NuA4 to act on specific genes by binding to 'activator proteins' that are found on the DNA. Tra1 is one of the biggest proteins in the cell, but its size makes it difficult to study and until now, its structure was unknown.

To determine the structure of Tra1, Díaz-Santín et al. extracted the protein from baker's yeast, and examined it using electron microscopy. The structure of Tra1 resembled a diamond ring with multiple protein domains that correspond to a band, setting and a centre stone. The structure was detailed enough so that Díaz-Santín et al. could locate various mutations that affect the role of Tra1. These locations are likely to be direct interfaces to the 'activator proteins'. Moreover, the study showed that Tra1 was similar to another protein that repairs damaged DNA.

These results suggest that Tra1 not only works as an activator target, but may also have a role in repairing damaged DNA, and might even connect these two processes. Yeast Tra1 is also very similar to its human counterpart, which has been shown to stimulate cells to become cancerous. Further studies based on these results may help us understand how cancer begins.
DOI: https://doi.org/10.7554/eLife.28384.002

*McMahon et al., 1998*; *Saleh et al., 1998*; *Vassilev et al., 1998*), an essential and highly conserved 433 KDa protein (*Figure 1—figure supplement 1*) that belongs to the Phosphoinositide 3-Kinase-related kinase (PIKK) family of cellular regulators, which includes mTOR, DNA-PKcs, ATM/Tel1, ATR/Mec1 and SMG-1 (*Baretić and Williams, 2014*; *Lempiäinen et al., 2009*). PIKKs are protein kinases that have diverse regulatory functions in transcriptional regulation, DNA repair, cell growth, metabolic control and mRNA surveillance but Tra1 is the only member that is catalytically inactive, due to loss of the DFG motif within the kinase active site (*Saleh et al., 1998*) (*Figure 1—figure supplement 2*). Although lacking catalytic activity, Tra1 is critical for coactivator function as it is a direct target for multiple activators (*Brown et al., 2001*) and enables the activities of SAGA and NuA4 to be directed at specific genes in order to stimulate their expression.

Activators contain transactivation domains (TADs) which directly target coactivators (*Näär et al., 2001*; *Ptashne, 1988*). Understanding the molecular mechanisms of TAD-coactivator interactions is a major challenge as TADs are poorly conserved, are often promiscuous and exhibit a strong compositional bias toward acidic, proline, glutamine or serine residues (*Mitchell and Tjian, 1989*), resulting in an intrinsically disordered protein region unless bound to a coactivator target (*Dyson and Wright, 2005*). In *S. cerevisiae*, activators such as VP16, Gal4, Gcn4 and Hap4 directly target Tra1 in vitro (*Brown et al., 2001*; *Fishburn et al., 2005*; *Herbig et al., 2010*; *Knutson and Hahn, 2011*; *Reeves and Hahn, 2005*) and in vivo (*Bhaumik and Green, 2001*; *Bhaumik et al., 2004*; *Larschan and Winston, 2001*; *Lin et al., 2012*), and the human homolog TRRAP interacts with the transcription factors c-Myc, E2F and E1A and is required for their stimulation of oncogenesis (*Ard et al., 2002*; *Bouchard et al., 2001*; *Deleu et al., 2001*; *Kulesza et al., 2002*; *McMahon et al., 1998*; *2000*), making Tra1/TRRAP a conserved activator target in all eukaryotes.

Mutations of Tra1 have been described that affect HAT activity without affecting coactivator complex integrity, indicating roles beyond activator targeting (*Knutson and Hahn, 2011*; *Mutiu et al., 2007*). Tra1 is also present in other chromatin-related complexes, including the SAGA-related complex SLIK (*Pray-Grant et al., 2002*) and the ASTRA complex (*Shevchenko et al., 2008*) from yeast. Similarly, TRRAP is present in four different human coactivator complexes STAGA, TFTC, PCAF and Tip60 (*Murr et al., 2007*). Interestingly, Schizosaccharomyces pombe contains two Tra1 paralogs which separately associate with SAGA and NuA4 (*Helmlinger et al., 2011*). Both Tra1 and TRRAP

are essential proteins; Tra1 is the only essential subunit of the SAGA complex (*Saleh et al., 1998*) and TRRAP disruption leads to early embryonic lethality in mice (*Herceg et al., 2001*). The high level of conservation of Tra1 sequence and function from yeast to human, its requirement for cellular viability and its presence in multiple coactivator complexes highlights its pivotal role in regulating gene expression. However, the molecular mechanisms behind Tra1 function are poorly understood, and the reason for its common integration within multiple complexes is unclear. To elucidate these aspects of its function, we determined an atomic structure of the Tra1 protein.

## Results

### Cryo-EM structure of *S. cerevisiae* Tra1

We over-expressed and purified *S. cerevisiae* Tra1 from its native host and determined its structure by single-particle cryo-EM to a resolution of 3.7 Å (*Figure 1—figure supplement 3*). Sidechains were visible throughout the reconstruction (*Figure 1—figure supplement 4A*) and an atomic model was built entirely de novo with 3474 residues (93%) assigned with excellent stereochemistry (*Table 1*), representing the first atomic structure for this member of the PIKK family. 270 residues were not resolved in the reconstruction, distributed across 15 chain breaks that are predicted to contain either loops or disordered protein. Tra1 has the domain structure characteristic of PIKK family proteins, consisting of HEAT, FAT, FRB, Kinase and FATC domains arranged sequentially from N- to C- terminus (*Figure 1A*) (*Baretić and Williams, 2014*; *Lempiäinen et al., 2009*). Alpha-helical solenoid repeats account for 86% of its mass which are contributed by the HEAT and FAT domains, and contain 49 HEAT repeats (labelled H1-H49) and 15 TPR repeats (labelled T1-T15) respectively (*Figure 1B,D* and *Video 1*). These are followed by FRB, kinase and FATC domains at the C-terminus.

Tra1 resembles a diamond ring, where the HEAT domain forms the ring, the FAT and FRB domains combine to form the setting, and the kinase and FATC domains represent the centre stone

**Table 1.** Structure determination and refinement details.

| Data collection | |
| --- | --- |
| Particles | 182,285 |
| Pixel Size (Å) | 1.06 |
| Defocus Range (-µm) | 1.5–3.5 |
| Voltage (kV) | 300 |
| Electron Dose (e- Å$^{-2}$) | 44 |
| **Refinement and validation** | |
| Resolution (Å) | 3.7 |
| Map CC (whole unit cell) | 0.817 |
| Average B-factor (Å$^2$) | 79.2 |
| RMS deviations – Bonds (Å) | 0.010 |
| RMS deviations – Angles (deg) | 1.41 |
| EMRinger score | 1.62 |
| Molprobity Score | 2.29 |
| Clashscore | 8.27 |
| Ramachandran plot (%) | |
| Favoured | 84.6 |
| Allowed | 14.8 |
| Outlier | 0.64 |
| C-beta deviations | 0 |
| Rotamer Outliers (%) | 1.72 |

DOI: https://doi.org/10.7554/eLife.28384.008

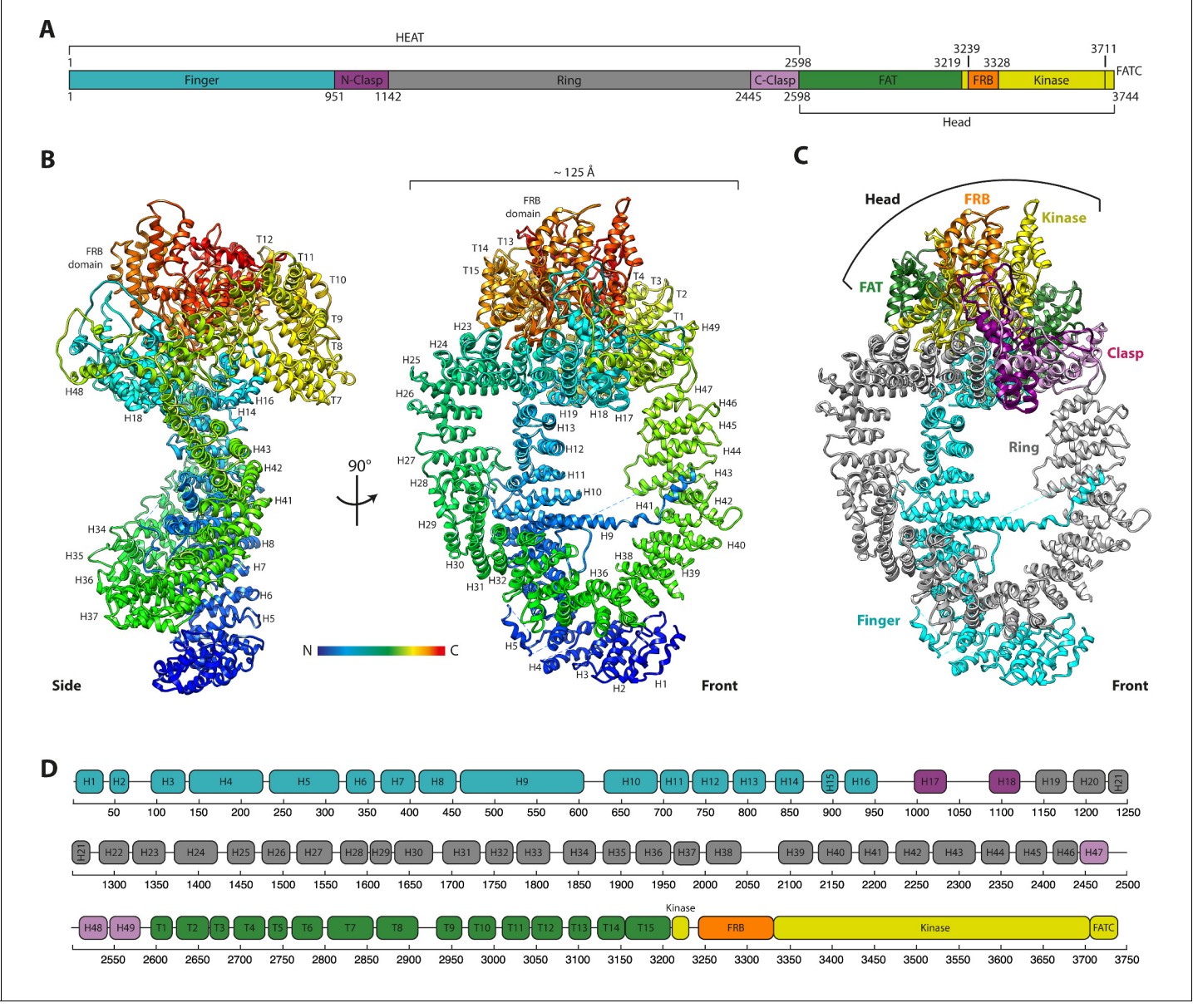

**Figure 1.** The cryo-EM structure of the Tra1 protein. (**A**) The domain organisation of Tra1. Colouring is matched to **Figure 1C,D**. The HEAT domain contains 49 HEAT repeats and the FAT domain (named after FRAP, ATM and TRRAP) contains 15 TPR (tetratricopeptide) repeats. The FRB domain (FKBP-Rapamycin-Binding), Kinase and FATC (FRAP-ATM-TRRAP C-terminus) domains are located at the C-terminus. (**B**) Front and side views of the Tra1 protein. The N-terminus is represented in blue and transitions to red at the C-terminus. Missing residues are shown as dotted lines. Where visible, HEAT repeats H1-H49 and TPR repeats T1-T15 are labelled sequentially, from N- to C-terminus. (**C**) The ring organisation of the Tra1 protein defined by its topological regions of Finger, Clasp, Ring, FAT, FRB and Kinase. The two halves of the clasp are shown in different shades of purple. The view is from the front as in **Figure 1A**. (**D**) Schematic detailing the primary sequence positions of alpha solenoid repeats H1-H49 and T1-T15 and their correspondence to the regions defined in panels A and C.

DOI: https://doi.org/10.7554/eLife.28384.003

The following figure supplements are available for figure 1:

**Figure supplement 1.** Conservation plot and example of alignment.
DOI: https://doi.org/10.7554/eLife.28384.004

**Figure supplement 2.** Sequence alignment of FRB and Kinase domains from ATM, DNA-PKcs, mTOR and Tra1.
DOI: https://doi.org/10.7554/eLife.28384.005

**Figure supplement 3.** Purification, data collection, image processing and overall reconstruction of Tra1.
DOI: https://doi.org/10.7554/eLife.28384.006

**Figure supplement 4.** Resolution map and exemplary electron density for Tra1.

*Figure 1 continued on next page*

*Figure 1 continued*

DOI: https://doi.org/10.7554/eLife.28384.007

(*Figure 1C* and *Video 1*). Using that analogy, Tra1 can be broadly separated into four topological regions which we have termed Finger, Ring, Clasp and Head (*Figure 1A,C*). The Finger, Ring and Clasp regions lie within the HEAT domain, whereas the Head region contains the FAT, FRB, Kinase and FATC domains (*Figure 1A*). The Tra1 Head is therefore analogous to the Head or FATKIN (<u>FAT</u> plus <u>KIN</u>ase) regions defined for structures of ATM, DNA-PKcs, Mec1, mTOR/Tor, Tel1, and SMG-1 (*Aylett et al., 2016*; *Baretić et al., 2016*; *2017*; *Lau et al., 2016*; *Melero et al., 2014*; *Rivera-Calzada et al., 2005*; *Sawicka et al., 2016*) in that they all encompass the FAT, FRB, Kinase and FATC domains and represent the most structurally conserved feature amongst PIKK family members (*Figure 5—figure supplement 1*).

## The HEAT domain forms two distinct alpha solenoids

The HEAT domain begins with the Finger, which consists of an alpha solenoid formed of N-terminal HEAT repeats H1-H16 (*Figure 1B and C*), and is equivalent to the 'Spiral' region of mTOR/Tor and ATM, or the Arm/Bridge region of DNA-PKcs (*Sharif et al., 2017*) (*Figure 5—figure supplement 1*). Finger Repeats H1-H6 form a flap over the midpoint of the Ring and appear flexible, as suggested by local resolution analysis (*Figure 1—figure supplement 4B*), and continues through H7-H16 which runs across the Ring toward the Head. H9 is an unusually large HEAT repeat and contains a 99-residue insertion (residues 482 to 580) between its helices (*Knutson and Hahn, 2011*). Two-thirds of this insertion was resolved in the reconstruction and is an unusual feature, as the N-terminal helix of H9 extends across the Ring to contact the opposite side at H42-H43 (*Figure 1B*). This interaction is corroborated by BS3-crosslinking experiments on the complete SAGA complex (*Han et al., 2014*), but the function of the H9 insertion is unclear, given that it is poorly conserved (*Figure 1—figure supplement 1*) and is not essential for viability (*Knutson and Hahn, 2011*).

After H16, the Finger solenoid is terminated by a 38-residue loop (residues 960–996) containing a two-stranded beta sheet (*Figure 1—figure supplement 4A*) and a second solenoid is formed by repeats H17-H49. This is the largest continuous solenoid in Tra1 and dominates the appearance of Tra1, forming a large closed ring approximately 125 Å in diameter. This solenoid starts with the N-clasp (H17-H18), continues with the Ring region (H19-H46) and ends with the C-clasp (H47-H49) which abuts the N-Clasp to close the ring (*Figure 1C*). The Clasp contains a significant proportion of insertions between its repeats, as predicted by sequence analysis (*Knutson and Hahn, 2011*), which form a set of interlocking loops that fix the Ring closed (*Figure 2A*). The Ring has a cradle-like conformation (*Figure 1B*) and its juxtaposition with the Finger creates large solvent-accessible channels between them, creating a highly open conformation and a large surface area (*Figure 1C*). As well as closing the Ring, the Clasp is also partly continuous with the FAT domain solenoid, in effect creating a 'figure-of-eight' conformation (*Figure 2—figure supplement 1*). Collectively, the

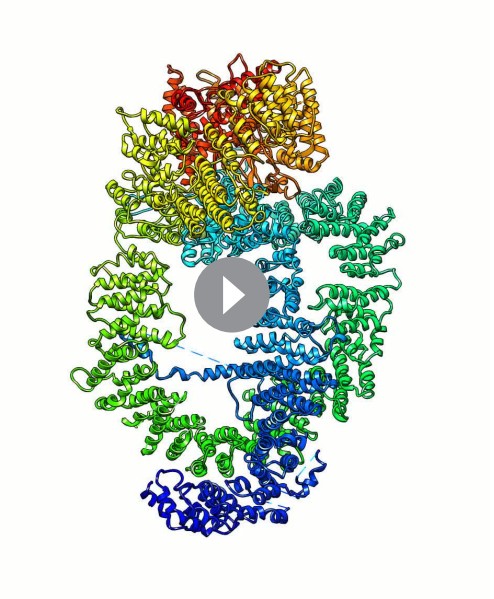

**Video 1.** The structure of the Tra1 protein and its position within the SAGA complex. A rotating movie of the Tra1 protein is shown, coloured from blue at the N-terminus to red at the C-terminus. The colours then transition to those defined for the regions described in *Figure 1C* and are labelled within the movie. Finally, the view zooms out and shows the fit of Tra1 within a reconstruction of the SAGA complex.

DOI: https://doi.org/10.7554/eLife.28384.009

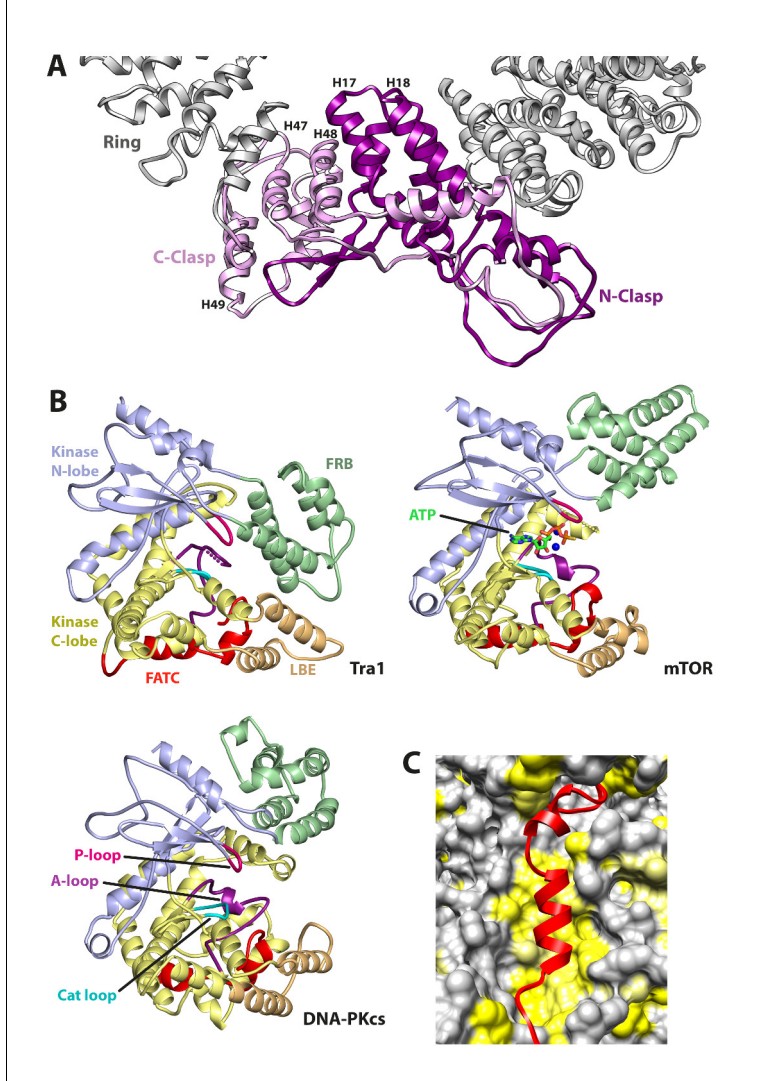

**Figure 2.** Structural features of Tra1. (**A**) The clasp region of Tra1 is shown as part of the ring. Repeats H17, H18, H47, H48 and H49 form the clasp and contain interlocking protein loops that fix the clasp together. The view is from the front. Finger and Head regions have been removed for clarity. (**B**) Comparison of FRB, Kinase, LBE and FATC domains between PIKK family members. The kinase domain is split into N-lobe and C-lobe halves. Structures of Tra1, mTOR with bound ATP (PDB code 4JSP) and DNA-PKcs (PDB code 5LUQ) are indicated. The phosphate binding loop (P-loop), catalytic loop (Cat loop) and Activation loop (A-loop) are highlighted for each structure. (**C**) The FATC domain binds to a hydrophobic pocket. FATC is shown as a red riboon, bound to the Kinase domain shown in surface representation. Hydrophobic surfaces are coloured in yellow.

DOI: https://doi.org/10.7554/eLife.28384.010

The following figure supplement is available for figure 2:

**Figure supplement 1.** Figure of eight organisation of the Ring (grey), Clasp (purple, FAT (green) and FRB (orange) domains.

DOI: https://doi.org/10.7554/eLife.28384.011

N-Clasp, Ring and C-Clasp regions are topologically equivalent to the 'Bridge', 'Railing' and 'Cap' regions defined for the HEAT domain of Tor (*Baretić et al., 2016*) and ATM (*Baretić et al., 2017*), in that they separate the N-terminal Finger/Spiral from the C-terminal FAT domain, although their relative positioning is different.

## The head contains an inactive kinase domain

The FAT, FRB, Kinase and FATC domains combine to form the Head (*Figure 1A and C*). The FAT domain contains 15 TPR repeats (T1-T15) and together with the FRB domain, surround the Kinase and FATC domains. The kinase domain has a fold typical for PIKK and PI3K catalytic domains, superposing onto the mTOR (*Yang et al., 2013*) and DNA-PKcs (*Sibanda et al., 2017*) kinase domains with RMSDs (Cα) of 2.7 Å and 2.9 Å respectively. However, although the relative positions of the catalytic, activation and phosphate-binding loops of mTOR/DNA-PKcs are preserved in Tra1 (*Figure 2B*), the critical residues required for ATP/Mg binding and catalysis are not conserved, and the Tra1 activation loop contains an 18-residue insertion compared to its counterparts in the catalytic PIKKs (*Figure 1—figure supplement 2*). The relative juxtaposition of FRB and kinase domains also differ as the DNA-PKcs and mTOR FRB domains are positioned away from the active site cleft, whereas the Tra1 FRB domain occludes it, contacting the LBE (mLST8-Binding-Element) on the opposite site of the cleft (*Figure 2B*). These conformational differences between DNA-PKcs/mTOR and Tra1 likely reflect that Tra1 is a pseudokinase, allowing the divergence of its catalytic features.

The FATC domain is integral to the kinase domain and is sandwiched between the LBE and the Kinase C-terminal lobe, forming a plug over a large hydrophobic cavity within the kinase domain (*Figure 2C*). Disruption of this plug is likely to destabilise the kinase domain significantly and/or induce conformational changes in adjacent domains, explaining why mutations within FATC often result in loss of viability or decreased stability of Tra1 (*Hoke et al., 2010*). Although the FATC domain is hypothesized to be critical for regulating catalytic activity of PIKKs (*Yang et al., 2013*), its sensitivity to mutagenesis within the kinase-inactive Tra1 and its protection of the hydrophobic cavity from solvent suggests it also has a key role in maintaining structural integrity.

## Tra1 occupies a peripheral position within the SAGA complex

Given the common presence of Tra1 in SAGA and NuA4, a key question is how Tra1 is incorporated into each coactivator, and whether complex integration results in functional or mechanistic differences. To examine its interactions with the SAGA complex, Tra1 was fitted into a recent 30 Å negative stain EM reconstruction of wild-type *S. cerevisiae* SAGA (EMD-2693) which exhibits a bilobal structure (*Durand et al., 2014*). A unambiguous fit was found within 'Lobe A' (*Figure 3* and *Video 1*) and indicate that no gross conformational changes are required to fit Tra1 into this SAGA reconstruction. The remaining SAGA density is contained within the crescent shaped 'Lobe B' which accounts for the remaining SAGA subunits, hence the contact between lobes A and B represents a major interface between Tra1 and SAGA. However, the interface is small (*Figure 3* and *Video 1*), demonstrating that Tra1 occupies a peripheral position within SAGA (*Setiaputra et al., 2015*; *Han et al., 2014*; *Wu et al., 2004*) and is not required for its structural integrity as a scaffolding platform (*Helmlinger et al., 2011*; *Wu and Winston, 2002*). The interface is localized to one side of of Tra1, primarily around the FAT domain at TPR repeats T1-T7 (residues 2572–2830) but also at the C-terminal half of the Ring at repeats H41-H44 (residues 2150–2350) (*Figures 1* and *3*), which clearly represent sites of intermolecular contact between Tra1 and the remaining SAGA subunits. This is supported by BS3-crosslinking experiments (*Setiaputra et al., 2015*; *Han et al., 2014*), which detected five Tra1 residues (K2351, K2713, K2781, K2808 and K2815) that lie adjacent to this interface, making intermolecular crosslinks to subunits Taf12, Spt20, Ada1 and Sgf73 (*Figure 3*). Given that Taf12 is the most frequently identified crosslinking partner of Tra1 (accounting for 6/13 intermolecular crosslinks), we suggest that it lies within or close to the observed interface, which is consistent with an earlier proposed model of the SAGA complex determined by negative stain EM, albeit at lower resolution (*Wu et al., 2004*). Similarly, as Ada1 can form a heterodimer with Taf12 (*Selleck et al., 2001*), and its deletion causes the release of Tra1 from SAGA (*Wu and Winston, 2002*), it is also likely to lie close to this interface. The BS3-crosslinking experiments also detected three additional residues that make intermolecular crosslinks to Spt3, Sgf73 and Taf12, but are located distal from the observed interface, being located on the Finger (K476) or on the opposite side of the FAT domain (K3161 and K3175) (*Figure 3*). Although the identification of crosslinked amino acids can suffer from false positives, the position of these residues away from the main interface are not necessarily inconsistent with forming intermolecular contacts, as elements of SAGA that are less globular in structure and are poorly resolved by negative stain EM may project away from Lobe B to make contacts with Tra1, such as extended loops or helices.

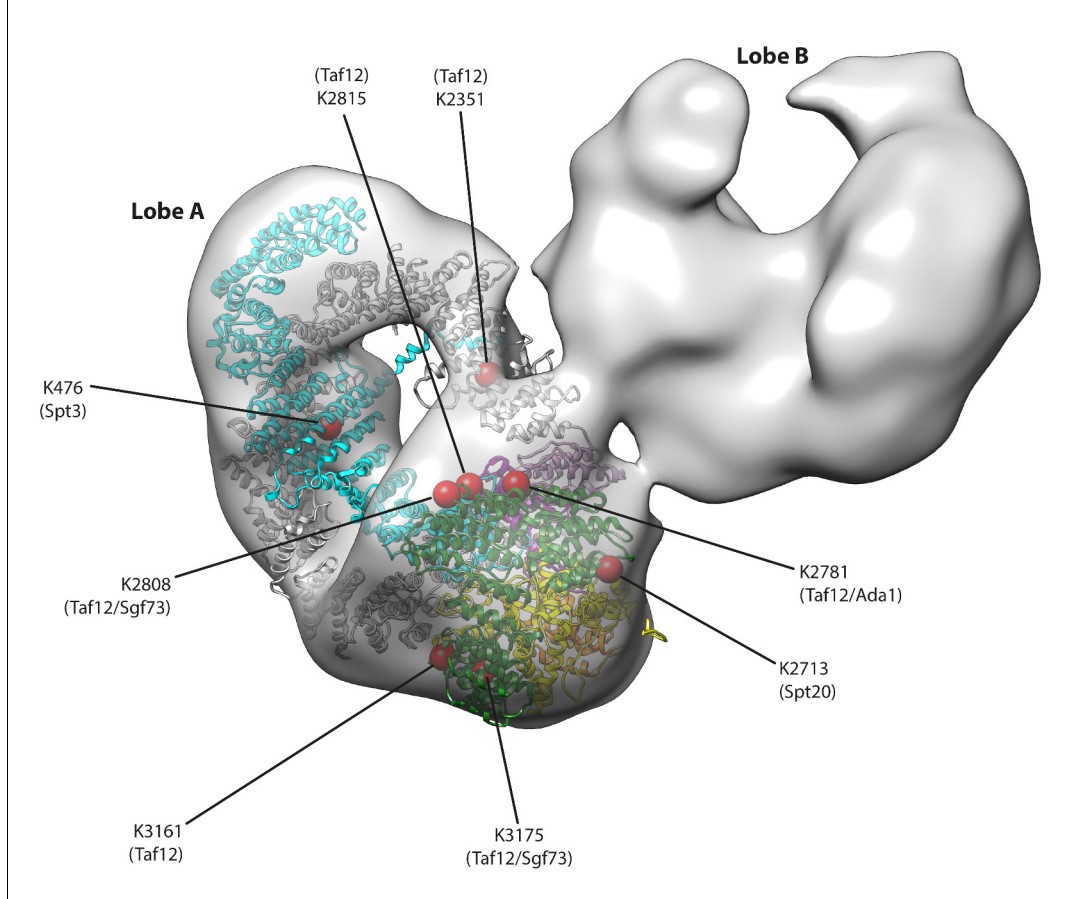

**Figure 3.** Tra1 occupies a peripheral position within SAGA. A negative stain reconstruction of S. cerevisiae SAGA was retrieved from the EMDB (EMD-2693) and Tra1 was fitted into the map using Chimera. An fit to the density was found to lobe A, and displayed as a ribbon model within the SAGA map (coloured as in *Figure 1C*). Eight red spheres on the Tra1 model indicate crosslinking sites to SAGA determined by mass spectrometry, and are labelled with residue position and target subunit within SAGA. Also see *Video 1*.

DOI: https://doi.org/10.7554/eLife.28384.012

The following figure supplement is available for figure 3:

**Figure supplement 1.** Representative 2D class average of Tra1 (left panel) was filtered to 21 Å (middle panel) for comparison with 2D class average determined for the NuA4 complex (Chittuluru et al.

DOI: https://doi.org/10.7554/eLife.28384.013

## Mutations mapped to the Tra1 structure reveal activator binding sites

To reveal potential activator binding sites, mutations that disrupt targeting by activators VP16, Gcn4, Rap1, and Gal4 (*Brown et al., 2001*; *Knutson and Hahn, 2011*; *Lin et al., 2012*) were mapped to the structure (*Figure 4A*). Two Tra1 mutants defective for interaction with Gal4 contain five amino acid substitutions (H400Y and D397N/S404F/D469N/V1115I) (*Lin et al., 2012*), four of which cluster at repeats H7-H8 within the N-terminal half of the Finger and are solvent exposed, indicating a binding site for the Gal4 activator (*Figure 4A*). Importantly, these mutations are highly specific for Gal4, and do not appear to affect interaction with other activators. Similarly, two deletion mutants of Tra1 (Δ88–165 and Δ319–399, located at H3 and H6-H7 respectively) disrupt coactivator recruitment by activators Gcn4 and Rap1 (*Figure 4*) but do not disrupt recruitment by Gal4 (*Knutson and Hahn, 2011*). These mutations are all located in the N-terminal half of the Finger but are specific for their affected activators, suggesting that the Finger contains multiple but distinct binding sites for different activators. Interestingly, the N-terminal half of the Finger contacts the Ring at repeats H31-H38 (*Figure 1B*), which was found to mediate interactions between human TRRAP and c-Myc, specifically within repeats H36-H38 (*Park et al., 2001*), and suggest that this pole of Tra1 (i.e. opposite to the Head) is generally targeted by activators. The juxtaposition between

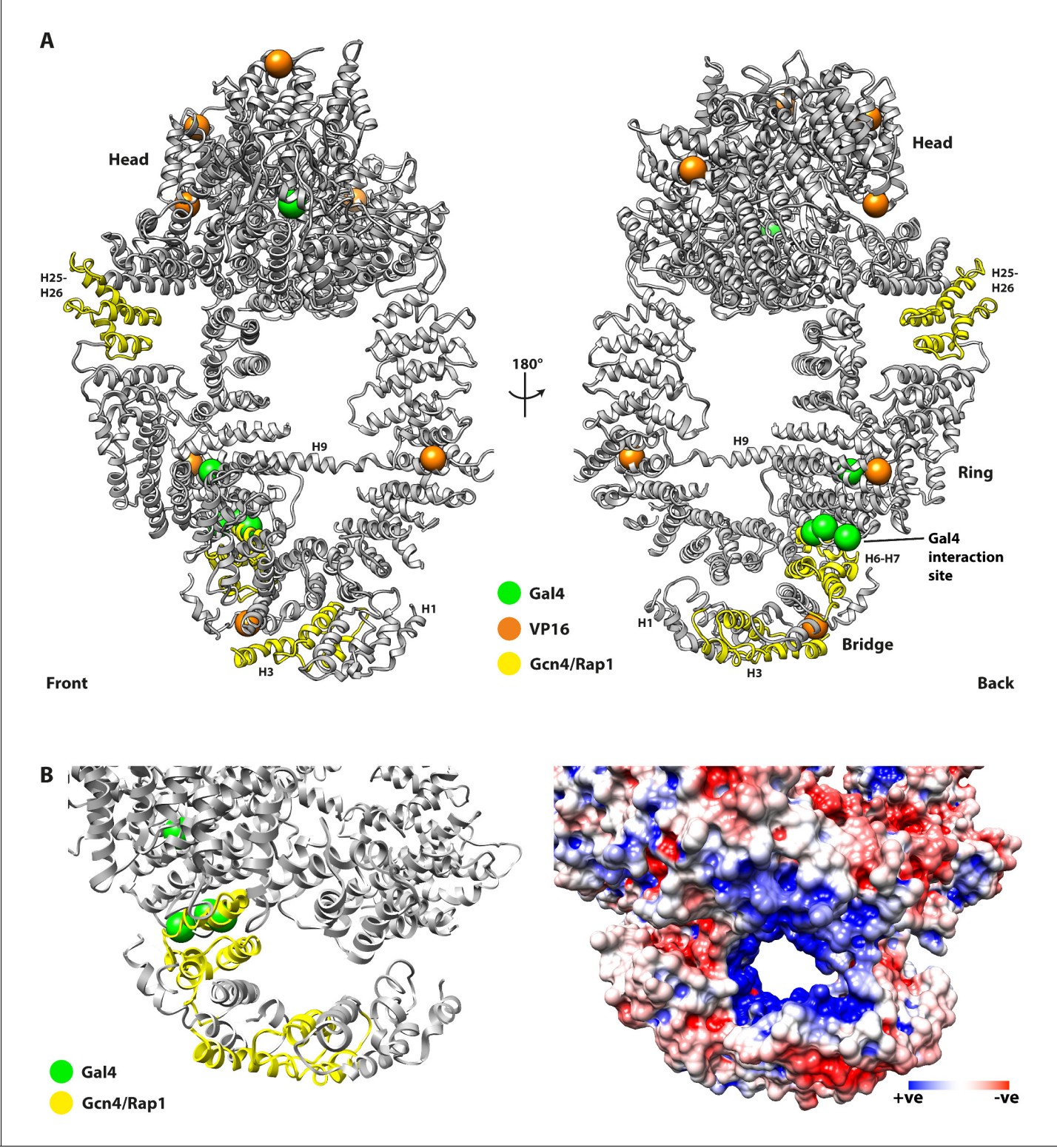

**Figure 4.** Mutations of Tra1 that disrupt activator targeting are distributed across the Tra1 structure. (**A**) Front and back view of Tra1 are shown together with mutations that disrupt activator targeting of SAGA/NuA4. Tra1 is shown as grey ribbon, and locations of amino acid substitutions and deletions are shown as spheres and yellow ribbon respectively. (**B**) Left panel shows a close up view of the N-terminal region of the Finger and its position relative to the Ring, with mutations that impair activator recruitment coloured as in *Figure 4*. Right panel has the same view but showing an

*Figure 4 continued on next page*

*Figure 4 continued*

electrostatic surface potential, highlighting the negatively charged channel that is lined by the Finger and Ring regions. Electrostatic surface potentials were calculated using PDB2PQR (*Dolinsky et al., 2007*) and APBS (*Baker et al., 2001*) tools implemented in Chimera.

DOI: https://doi.org/10.7554/eLife.28384.014

Finger and Ring in this region also forms a channel lined by positively charged residues contributed by both Finger and Ring (*Figure 4B*), which may assist binding of the acidic transactivation domains frequently found in activators that target Tra1. However, activators may also be targeted to other regions of Tra1, as Gcn4 and Rap1 are disrupted by deletions in the Ring at H25-H26 (Δ1424–1508) (*Knutson and Hahn, 2011*), mutations that disrupt VP16 are clustered around the Head (*Figure 4A*), and in vitro experiments show VP16 interacts with the C-terminal regions of Tra1 (*Brown et al., 2001*).

## The conformation of Tra1 is strikingly similar to DNA-PKcs

Comparison of Tra1 to other PIKK structures show that mTOR (*Aylett et al., 2016*; *Yang et al., 2013*) (*Figure 5—figure supplement 1*) and ATM (*Wang et al., 2016*) have a similar arrangement of FAT, Kinase and FATC domains but the conformation of their HEAT domains differ significantly. This is unsurprising given that PIKKs have highly divergent functions, and typically form complexes with a diverse range of regulatory factors which often target the HEAT domain (*Aylett et al., 2016*; *Baretić and Williams, 2014*; *Spagnolo et al., 2012*). However, the entire Tra1 structure is strikingly similar to human DNA-PKcs (*Sibanda et al., 2017*) (*Figure 5*) an essential DNA double strand break (DSB) repair factor. Despite only having having 18% sequence identity, both Tra1 and DNA-PKcs have similar 'diamond ring' topologies, and regions analogous to Finger, Clasp, Ring and Head can be defined for DNA-PKcs, resulting in the same relative positioning as Tra1 (*Figure 5*). The largest difference in conformation is between the Tra1 Finger, which is equivalent to the 'N-terminal Unit' (*Sibanda et al., 2017*) or 'Arm/Bridge' region (*Sharif et al., 2017*) defined for DNA-PKcs. Specifically, the N-terminal repeats of the Finger region in Tra1 form a flap over the Ring, whereas the equivalent zone DNA-PKcs forms an arch whose concave surface was hypothesized to be a DNA-binding site required for synapsis of a DSB within a DNA-PKcs dimer (*Sibanda et al., 2017*). Although the equivalent region of Tra1 does not form an arch and cannot sterically accommodate duplex DNA, these repeats contain a highly positively charged surface (*Figure 4B*) and local resolution analysis suggests they are flexible (*Figure 1—figure supplement 4B*), potentially indicating the presence of a nucleic acid binding site.

## Discussion

Coactivators are far less numerous than activators, and represent hubs within transcriptional regulation. Understanding the molecular mechanisms behind coactivator function is essential for elucidating how complex programmes of gene expression are established. Although SAGA and NuA4 are well characterized in terms of their enzymatic activities and genome localization, their interactions with activators are poorly understood. Tra1 has been identified as a key activator target, but the molecular details of its interaction with activators and its parent complexes are yet to be determined. To provide insight into this aspect of coactivator function, we determined the structure of Tra1 by cryo-EM, and built an atomic model of the complete protein, allowing activator-disrupting mutations to be mapped to the structure and highlighting potential TAD binding sites. The model also fitted unambiguously into an existing reconstruction of SAGA to reveal its binding interface and integration within the complex.

The integration of Tra1 within SAGA leaves Tra1 relatively unimpeded for activator binding, as its interaction occludes very little of its solvent accessible surface. Similarly, as its conformation within SAGA appears unchanged from apo-Tra1 (*Video 1*), this suggests that activators cannot discriminate SAGA from NuA4 via Tra1 targeting alone, and other subunits specific to SAGA may act as additional activator targets and provide specificity e.g. numerous activators can directly target the SAGA subunits Gcn5, Ada1, Taf6 and Taf12 (*Klein et al., 2003*; *Reeves and Hahn, 2005*; *Zhang et al., 2014*). It is possible that the presentation of Tra1 by NuA4 could restrict or alter its binding to

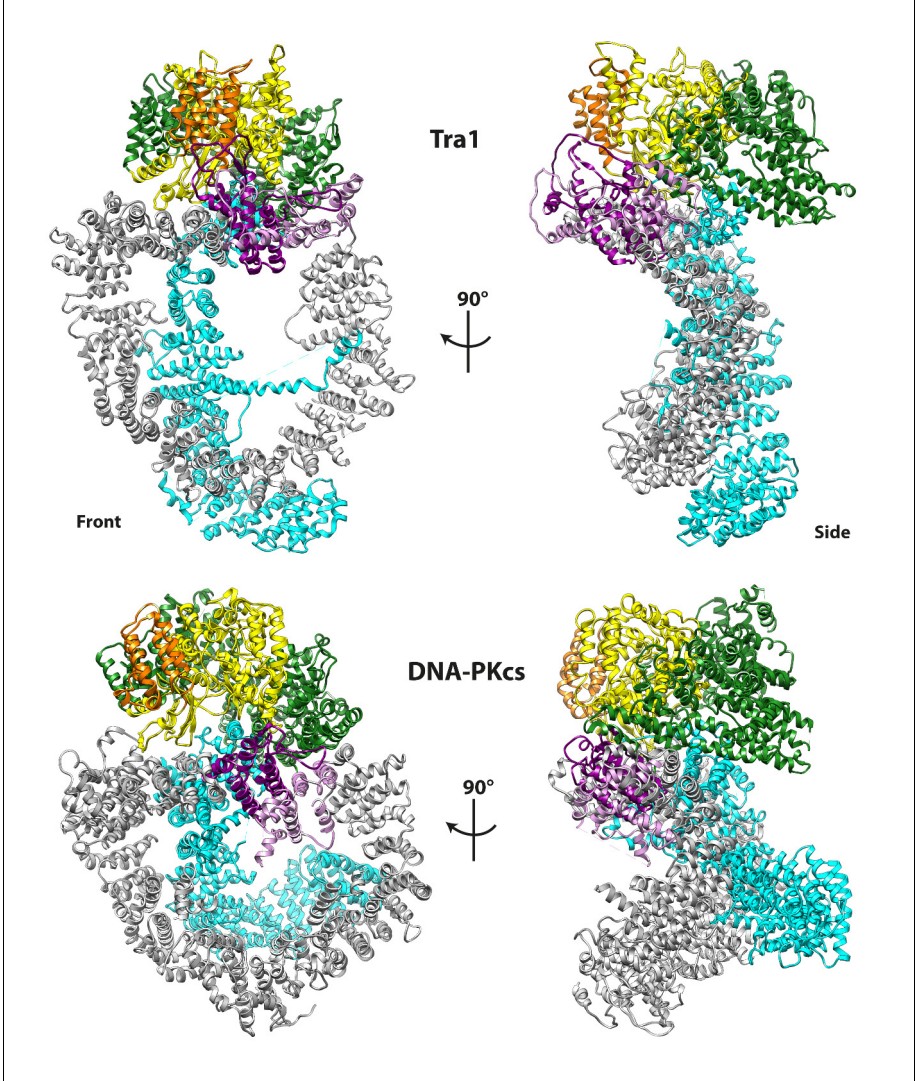

**Figure 5.** Tra1 is structurally homologous to DNA-PKcs. DNA-PKcs (PDB code 5LUQ) was superposed with Tra1, and regions of DNA-PK analogous to Finger, Ring, Clasp, FAT, FRB and Kinase are coloured according to the scheme given in *Figure 1C and D*. Front and side views are shown and highlight their similar topology.

DOI: https://doi.org/10.7554/eLife.28384.015

The following figure supplement is available for figure 5:

**Figure supplement 1.** HEAT domain comparison with mTOR, ATM and DNA-PKcs.

DOI: https://doi.org/10.7554/eLife.28384.016

activators in comparison to SAGA, but this remains to be determined; visual comparisons of our Tra1 2D class averages with those from a previously determined cryo-EM reconstruction of NuA4 (*Chittuluru et al., 2011*) (*Figure 3—figure supplement 1*) shows that the entirety of NuA4 closely matches Tra1 in appearance, indicating that the remaining NuA4 subunits are highly dynamic and/or have dissociated in the reconstruction. As endogenous purifications of NuA4 do not co-purify subunits of SAGA (and vice versa), Tra1 is unlikely to connect the two coactivators into a single complex and its presence in separate coactivators may result from overlapping contact sites with both SAGA and NuA4, precluding their assembly around the same molecule of Tra1. Hence the SAGA contact sites identified in Tra1 may also be exploited for NuA4 interactions which is supported by deletion mutations within Tra1 that simultaneously cause defects in SAGA and NuA4 assembly (*Knutson and Hahn, 2011*).

Mutations that disrupt the interaction of Tra1 with Gal4, Gcn4, Rap1 and VP16 are spread across Tra1, located within the Finger, Ring and Head regions, and are predominantly distal from the interface with SAGA. However, the precise mechanism of disruption remains unknown; as well as specifically abrogating a Tra1-activator interface, these mutations may also cause an allosteric change, affect protein stability or some combination thereof. Nevertheless, 4/5 of the amino acid substitutions that disrupt interaction with Gal4 are located in the Finger at solvent exposed sites, and three are spatially proximal to each other (D397N/H400Y/S404F), (*Figure 4*), strongly suggesting the presence of a direct Gal4 interface. Additionally, of the three deletion mutants that affect Gcn4 and Rap1, two are also located in the Finger adjacent to the Gal4 interface on its N-terminal side (H3 and H6-H7), hence we suggest that the Finger may function as a platform for interacting with multiple activators. The Finger is unlikely to be the only activator interaction site, given the location of the third deletion mutant in the Ring (proximal to the Head at H25-H26), the clustering of mutations that affect the interaction with VP16 within the Head, and an in vitro interaction between a C-terminal fragment of Tra1 and VP16 (*Brown et al., 2001*). A critical feature of these mutations is the ability to disrupt specific activators whilst leaving others unaffected i.e. mutations that disrupt the interaction of Gal4 with Tra1 do not affect Gcn4 (*Lin et al., 2012*) and vice versa (*Knutson and Hahn, 2011*). Conversely, the mutations that affect Gcn4 also effect Rap1, indicating that these activators target Tra1 in a similar manner. Although allosteric effects cannot be excluded, the simplest mechanism is that Tra1 harbors multiple interfaces for activators that can be specific for single activators (Gal4) or bind multiple activators (Gcn4 and Rap1). The presence of multiple interfaces would therefore provide a mechanism for Tra1 to integrate signals from activators, allowing multiple activators to cooperate in stimulating transcription. Individual binding sites are also likely to vary in their affinity and kinetics of interaction, further tuning the strength of transcriptional activation. More intricate mechanisms can also be hypothesized, such as competition between different activators for the same binding site, or by allosteric changes upon activator binding that alter its interaction with other activators and/or coactivator components. Although Tra1 required no conformational changes to fit into SAGA (*Figure 3*), HEAT repeat proteins are highly flexible (*Kappel et al., 2010*) and Tra1 may undergo conformational changes upon interacting with other factors such as activators. In that regard, the N-terminal part of the Finger is the most structurally dynamic part of Tra1 (*Figure 1—figure supplement 4B*) and makes extensive contacts with the Ring, so activator binding in this location may induce conformational changes and stimulate allosteric changes within Tra1 that may exert effects on its parent histone-modification complex.

The remarkable and unexpected structural homology to the DNA repair factor DNA-PKcs suggests that Tra1 has roles beyond transcriptional activation. Yeast lack a homolog for DNA-PKcs and the Tra1 parent complex NuA4 is required for double strand break (DSB) repair (*Bird et al., 2002*), so it is tempting to speculate that Tra1 may have functional similarities with DNA-PKcs and a direct role in DNA repair. DNA-PKcs mediates ligation of double strand breaks (DSBs) by forming a synaptic complex with Ku70-Ku80 and aligning the broken DNA ends, whereupon its kinase becomes active and coordinates further assembly of the repair machinery (*Dobbs et al., 2010*). Although Tra1 lacks the kinase activity that is crucial for DNA-PKcs function (*Chen et al., 2005*; *Cui et al., 2005*), the structural homology between Tra1 and DNA-PKcs suggests that Tra1 might retain the non-catalytic features of DNA-PKcs in binding nucleic acids and/or recruiting additional repair factors. Recruitment of active kinases may then substitute for lack of Tra1 catalytic activity, such as ATM which interacts with Tip60 (the human homolog of NuA4) in human cells (*Sun et al., 2005*). Although a direct role in DNA repair remains speculative, connections between Tra1 and DNA damage are already well established, as depletion of TRRAP compromises DSB repair (*Murr et al., 2006*; *Robert et al., 2006*) and its parent complex Tip60 is recruited to DSBs in a TRRAP-dependent manner (*Murr et al., 2006*), resulting in H4 acetylation that facilitates repair. TRRAP also forms a complex with the MRE11, RAD50, and NBS1 (MRN) complex (*Robert et al., 2006*), a key sensor of DSBs that also recruits PIKK family member ATM to sites of DSBs (*Falck et al., 2005*). In this manner, MRN could function analogously to Ku70-Ku80, which recruits DNA-PKcs to sites of DSBs. Additionally, NuA4 can recognize DSBs directly (*Bird et al., 2002*) and SAGA and NuA4 preferentially acetylate the ends of a linear chromatin template (*Vignali et al., 2000*), hence Tra1 may provide this recognition capability, given its homology to DNA-PKcs which has affinity for DNA ends (*Gottlieb and Jackson, 1993*).

As a direct target of multiple activators, and as common component of SAGA and NuA4, Tra1 is central to transcriptional regulation. However, its size and presence in additional chromatin-related complexes, and its homology to DNA-PKcs points to a role beyond activator targeting. The structure presented here is an important step toward discovering those roles, and further structural and biochemical studies of Tra1 bound to activators and/or its parent complexes will elucidate new mechanisms of its functions.

## Materials and methods

### Protein expression and purification

The *S. cerevisiae* Tra1 coding sequence (YHR099W) from genomic DNA was PCR amplified and cloned into a galactose-inducible pRS424-based expression vector (courtesy of K. Nagai, MRC LMB, Cambridge) with a N-terminal 3xFLAG tag. The plasmid was transformed into S. cerevisiae strain BCY123 (MATα pep4::HIS3 prb1::LEU2 bar1::HIS6 lys2::GAL1/10GAL4 can1 ade2 trp1 ura3 his3 leu23,112) and transformants selected on SC plates lacking tryptophan (Yeast Nitrogen Base, Trp dropout mix (Formedium Ltd., UK), 2% glucose, 50 mg/ml adenine) at 30°C for 2 days before making a glycerol stock for storage at −80°C. The following liquid shaker cultures were all made with the same media omitting agar, and incubated at 30°/185 rpm; a one litre pre-culture was prepared from the glycerol stock, and incubated overnight. The pre-culture was centrifuged and washed with sterile dH20 and used to inoculate a 24 litre expression culture with glucose replaced by 2% Raffinose to a starting OD of 0.1–0.2 and incubated until OD ~0.8. Tra1 expression was then induced with 2% Galactose for 6 hr before harvesting by centrifugation. Cells were frozen in liquid N2 for storage at −80°C.

Cells were thawed and resuspended in an equal volume of Buffer A (125 mM HEPES 8.0, 250 mM NaCl, 1.5 mM MgCl2, 10% glycerol, 0.1% IGEPAL CA-630, 0.5 mM DTT) supplemented with protease inhibitors (1.25x Roche cOmplete Ultra plus AEBSF (210 µM), Aprotinin (0.3 µM), Benzamidine (6.5 mM), Leupeptin (105 µM), E-64 (2.8 µM), PMSF (1.15 mM) and Pepstatin (200 µM)) for pipetting into liquid N2 and subsequent lysis by cryo-milling (SPEX 6870 freezer mill), followed by storage at −80°C. All following purification steps were completed at 4°C. Lysate powder was thawed, supplemented with Benzonase (1.5 µl/10 ml lysate, Sigma E8263) for 20 min, and sonicated before centrifugation (Ti45 rotor, 35000 rpm, 120 min). Clarified lysate was filtered and adjusted to pH 8.0 with 1M HEPES (pH 8.0) and incubated with 2 ml Anti-FLAG agarose (Sigma A2220) for 3 hr before washing with modified buffer A (as Buffer A but 50 mM HEPES pH 8.0, 0.3 mM DTT, protease inhibitors cocktail (Sigma S8830)) and elution with 0.5 mg/ml 3x FLAG peptide (Generon) in the same buffer. FLAG eluates were pooled and diluted to match conductivity of buffer B (50 mM HEPES 8.0, 100 mM NaCl, 1.5 mM MgCl2, 0.5 mM DTT) and applied to a MonoS 5/50 GL column (GE Healthcare) equilibrated in buffer B. The column was washed with buffer B before gradient elution to 1M NaCl in buffer B over 25 CV. Tra1 containing fractions were assayed by SDS-PAGE, and appeared in both flowthrough and elution fractions. These were pooled, diluted to match conductivity of buffer B and applied to a MonoQ 5/50 GL column (GE Healthcare) in the same manner. Tra1-containing elution fractions were pooled and spin concentrated (Amicon Ultra) for injection onto a Superose 6 10/300 GL column (GE Healthcare) equilibrated in buffer B containing 150 mM NaCl. Fractions containing monomeric Tra1 eluted at 14 ml (*Figure 1—figure supplement 3*) and were pooled and spin concentrated to 0.1 mg/ml (Amicon Ultra).

### Cryo-electron microscopy

Aliquots of the Tra1 preparation were placed on negatively glow discharged lacey grids with ultrathin carbon over holes (Agar Scientific, UK) and vitrified in liquid ethane using a Vitrobot Mark IV (FEI, USA). Blotting was carried out at 4°C and 94% humidity. Due to the low protein concentration two subsequent applications of Tra1 were required to achieve the desired protein density on grids. Each application was followed by 20 s waiting time, with a short 0.5 s blotting after first application and 5 s blotting after the second. Data were acquired using a Titan Krios microscope (FEI) operated at 300 keV and equipped an energy filter (Gatan GIF Quantum, USA). The images were collected with a post-GIF K2 Summit direct electron detector (Gatan) operating in counting mode at a nominal magnification of 130,000x, corresponding to 1.06 Å per physical pixel. An energy slit with a width of

20 eV was used during data collection. The dose rate on the specimen was set to 5.5 electrons per Å2 per s and a total dose of ~44 e/Å2 was fractionated over 32 frames. Data were collected using EPU software (FEI) with a nominal defocus range set from −1.5 μm to −3.5 μm.

## Image processing and model building

Unless otherwise stated, RELION 2.0 (*Scheres, 2012*) was used for all subsequent processing steps. MotionCor2 (*Zheng et al., 2017*) was used for patch-based motion correction of movie frames followed by CTFFIND4 (*Rohou and Grigorieff, 2015*) to estimate the contrast transfer function (CTF) parameters of the corrected micrographs. An initial subset of the data was processed with Gautomatch (*Urnavicius et al., 2015*), using an automatically generated Gaussian reference. After initial particle extraction and reference-free 2D classification, selected 2D classes were used as a template for further iterations of particle picking with Gautomatch, yielding 418,339 particles from 1733 micrographs. These were subjected to reference-free 2D classification, and particles contributing to the best 2D classes (*Figure 1—figure supplement 3D*) were selected for 3D refinement. A previously published 13 Å resolution cryo-EM density map of DNA-PKcs (EMD-1102) was low-pass filtered to 40 Å and used as initial reference for 3D refinement, and the resulting consensus model was used as a reference map for 3D classification. The best 3D class containing 182,285 particles (44% of total) was used to perform a 3D refinement run, resulting in a 3.9 Å map. Substitution of the particles contributing to this map by particles from dose-weighted images calculated by MotionCor2 provided a final reconstruction at 3.7 Å resolution after a last run of 3D refinement. Reported resolutions are based on gold-standard Fourier Shell Correlation (FSC) curves between independently refined half-maps, using the 0.143 criterion. The resulting maps from refinement were post-processed by RELION and sharpened by a negative B-factor using an automated procedure. The final map was highly detailed with clear density for strands, helices and loops (*Figure 1—figure supplement 4A*). Sidechains were resolved throughout the reconstruction, allowing de novo building of 3474/3744 residues. Model building was performed with Coot (*Emsley et al., 2010*) and assisted by secondary structure predictions from PSIPRED (*Jones, 1999*), JPRED3 (*Cole et al., 2008*), and also reported within (*Knutson and Hahn, 2011*). The abundance of helices amongst the solenoid repeats greatly assisted building and assignment of sequence register. Maps were B-factor sharpened with phenix. auto_sharpen (1.11.1–2575) (*Adams et al., 2010*) or filtered to 5 Å to provide extra guidance during model building. Local resolution variations were estimated within RELION. The model was refined with phenix.real_space_refine using secondary structure restraints. For cross-validation of the model, atomic positions were randomly perturbed by up to 0.5 Å to remove model bias from prior refinement against all the data, and subsequently refined against a single (unmasked) half-map using secondary structure restraints. The refined model was used for FSC calculations against the same half-map ($FSC_{work}$), the withheld half-map ($FSC_{free}$), and the combined map ($FSC_{total}$) to monitor for over-fitting (*Figure 1—figure supplement 3*). Refinement half-maps correspond to the same half-maps used during gold-standard FSC resolution estimation. Refinement/validation statistics are shown in table S1. The fit of the Tra1 structure into a SAGA reconstruction (EMD-2693) was performed with fitting tools implemented in Chimera (*Pettersen et al., 2004*), and assessed by correlation score and visual appearance. Figures were generated with Chimera and PyMOL (1.8, Schrödinger, LLC.).

## Acknowledgements

Yeast strain BCY123 and expression plasmid pRS424 were a kind gift from Kiyoshi Nagai, and Wojciech Galej provided advice on yeast expression. We thank our colleagues at ISMB/Birkbeck for their support and use of the EM infrastructure (funded by Wellcome Trust grant 079605/2/06/2). We thank Nathan Carr for helpful discussions, David Houldershaw for computational support, and Gabriel Waksman, Anthony Roberts, Giulia Zanetti and Kerstin Kinkelin for critiquing the manuscript. Preliminary imaging was performed at Birkbeck College and primary imaging was performed at the Electron Bio-Imaging Centre at Diamond Light Source (Oxfordshire, UK). We are grateful to Alistair Siebert (Diamond Light Source) and Giulia Zanetti for their support during data collection. This work was funded by a Wellcome Trust and Royal Society Sir Henry Dale Fellowship (102535/Z/13/Z), Royal Society Research Grant (RG140138) and UCL Excellence Fellowship awarded to ACMC. LMDS carried out cloning, protein expression, purification and EM image processing. NL prepared and optimised cryo grids and performed preliminary imaging. EA and ACMC assisted in protein expression.

ACMC built the model, supervised the project, and prepared the manuscript with LMDS and NL. Coordinates of Tra1 have been deposited with the protein data bank under accession number 5OJS. Electron Microscopy reconstructions have been deposited with the EMDB under accession number EMD-3824.

## Additional information

### Funding

| Funder | Grant reference number | Author |
|---|---|---|
| Wellcome | 102535/Z/13/Z | Alan CM Cheung |
| Royal Society | RG140138 | Alan CM Cheung |
| University College London | Excellence Fellowship | Alan CM Cheung |

The funders had no role in study design, data collection and interpretation, or the decision to submit the work for publication.

### Author contributions

Luis Miguel Díaz-Santín, Formal analysis, Investigation, Visualization, Methodology, Writing—original draft, Writing—review and editing, Carried out cloning, protein expression, purification and EM image processing; Natasha Lukoyanova, Resources, Formal analysis, Investigation, Methodology, Writing—original draft, Writing—review and editing, Prepared and optimised cryo grids and performed preliminary imaging; Emir Aciyan, Resources, Investigation, Assisted in protein expression; Alan CM Cheung, Conceptualization, Resources, Formal analysis, Supervision, Funding acquisition, Validation, Investigation, Visualization, Methodology, Writing—original draft, Project administration, Writing—review and editing, Built the model, supervised the project, and prepared the manuscript

### Author ORCIDs

Alan CM Cheung (iD) http://orcid.org/0000-0001-6430-5127

### Decision letter and Author response

Decision letter https://doi.org/10.7554/eLife.28384.022
Author response https://doi.org/10.7554/eLife.28384.023

## Additional files

### Supplementary files

• Transparent reporting form
DOI: https://doi.org/10.7554/eLife.28384.017

### Major datasets

The following datasets were generated:

| Author(s) | Year | Dataset title | Dataset URL | Database, license, and accessibility information |
|---|---|---|---|---|
| Luis Miguel Díaz-Santín, Natasha Lukoyanova, Emir Aciyan, Alan CM Cheung | 2017 | Electron Microscopy reconstructions | https://www.ebi.ac.uk/pdbe/emdb/EMD-3824 | Publicly available at the EMDataBank (accession no. EMD-3824) |
| Luis Miguel Díaz-Santín, Natasha Lukoyanova, Emir Aciyan, Alan CM Cheung | 2017 | Coordinates of Tra1 | http://www.rcsb.org/pdb/explore/explore.do?structureId=5OJS | Publicly available at the RCSB Protein Data Bank (accession no: 5OJS) |

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
