## [Decision Letter]

Thank you for submitting your article "Cryo-EM structure of the SAGA and NuA4 subunit Tra1 at 3.7 Å resolution" for consideration by *eLife*. Your article has been reviewed by two peer reviewers, and the evaluation has been overseen by a Reviewing Editor and John Kuriyan as the Senior Editor. The following individuals involved in review of your submission have agreed to reveal their identity: Michael R Green (Reviewer #2).

The reviewers have discussed the reviews with one another and the Reviewing Editor has drafted this decision to help you prepare a revised submission.

Summary:

The manuscript of Dáz Santín et al. describes the cryo-EM structure of Tra1, the yeast orthologue of TRRAP. This is unique among the PIKKs in that it is a pseudokinase rather than a kinase. This is an excellent and straightforward report of the first TRRAP structure at a resolution sufficient to interpret the path of the polypeptide. As would be expected from sequence similarity, the kinase domain is similar to other PIKKs, whose structures have been reported, mTOR, TOR, DNA-PKcs and ATM. An emerging picture is that the PIKKs have a variety of conformations for the vast regions outside the FAT and kinase domains. As these structures have been reported, a variety of names have been given to structural regions (domains). There is in general little sequence similarity in the regions, so this lack of structural similarity is not surprising. The physical features shown here that render the Tra1 kinase inactive are new and interesting, as is the proposed role for the FATC element in Tra1. What is also surprising in the current report is the striking structural similarity of Tra1 to DNA-PKcs. The authors propose that this may imply related functions in DNA double-strand break repair. This is an intriguing proposition, and the Tra1 structure will provide an excellent framework designing mutations that could test this hypothesis, however, this is beyond the scope of the current manuscript. Overall, the results showing the structure of the target of this transcriptional activator will be very valuable to the transcription and DNA repair fields. Pending the resolution of a few issues (detailed below), the work is appropriate for publication in *eLife*.

Major issues:

The structure determination appears to be of high quality, and technical aspects are generally adequately described. There is no detailed description of model building or refinement. It might be helpful to expand on this. Are side chains visible throughout the polypeptide? From the local resolution plot, this may not be the case. Are the authors confident of the assignment of the sequence register throughout the structure? Are there breaks in the density? If so, where are these breaks? Hopefully the authors have deposited a model with PDB that includes a residue assignment throughout, since they have made to indication of uncertainty.

Along these lines, the FSC plot in Figure 1—figure supplement 3 shows unusual features. This is likely leading to a slight overestimate of the resolution, which appears borne out the quality of the maps in Figure 1—figure supplement 4. Please address/comment on both issues.

Concerning the statement, "The model also fitted unambiguously into an existing reconstruction of SAGA to reveal its binding interface and integration within the complex". The work as written hints at this interface, but doesn't really make concrete, testable predictions. Based on the crosslinking data depicted in Figure 3, which parts of Tra1 and the rest of SAGA are touching?

Figure 3—figure supplement 1. It would be better to reproduce the pertinent image from Chittuluru et al. 2011 (with permission, of course), rather than just reference it. Alternatively, if a 3D EMDB model is available, it could be back-projected for comparison.

Figure 4. It is surprising that the mutations, which disrupt activator targeting, are so broadly distributed throughout Tra1. The authors should further elaborate in their Discussion as to why they think this is. One would suspect that some are specific, some allosteric, and some lead to general misfolding/instability. Do any mutations cross-disrupt targeting between different activators? Can the likely effects of the mutations be categorized?

---

## [Author Response]

*Major issues:*

*The structure determination appears to be of high quality, and technical aspects are generally adequately described. There is no detailed description of model building or refinement. It might be helpful to expand on this. Are side chains visible throughout the polypeptide? From the local resolution plot, this may not be the case. Are the authors confident of the assignment of the sequence register throughout the structure? Are there breaks in the density? If so, where are these breaks? Hopefully the authors have deposited a model with PDB that includes a residue assignment throughout, since they have made to indication of uncertainty.*

We have expanded our descriptions of model building and refinement to include more detail, both in the main text and the Materials and methods. Sidechains were visible for the majority of the polypeptide, and the map was highly detailed, allowing the structure to be built with confidence and has been submitted to the PDB along with maps to the EMDB, and codes have now been added to the manuscript. The structure is mostly composed of α helical repeats which greatly assisted building, especially when combined with secondary structure predictions, providing confidence in our register assignment. Additionally, helices from the solenoid repeats were often amphipathic, burying their hydrophobic sidechains into the solenoid core. This was a useful aid in building into regions of the reconstruction that were of lower quality. 270/3744 residues were not visible, spread over 15 chain breaks and we have added this information to the manuscript.

*Along these lines, the FSC plot in Figure 1—figure supplement 3 shows unusual features. This is likely leading to a slight overestimate of the resolution, which appears borne out the quality of the maps in Figure 1—figure supplement 4. Please address/comment on both issues.*

The FSC plot in Figure 1—figure supplement 3 was intended to represent the FSC of both masked (orange) and unmasked (green) reconstructions. However, the data was misplotted from our data file. We apologise for the mistake and have replaced both traces with masked and unmasked FSC with the correct values, resulting in a more canonical appearance. The resolution limit (defined by the cutoff at 0.143) is unchanged at 3.7 Å. From our experience with crystal structure determination at low resolution (e.g. PDB codes 4A3M, 4A3I, 4A3J, 3HOX, 3HOZ (3.7-3.9 Å)), we believe the features in the map are representative for this resolution estimate. We fear that Figure 1—figure supplement 4 may not have fully conveyed the quality of the reconstruction, as we deliberately used β strand containing regions that are typically the most challenging to resolve. We have enlarged the existing panels and included two extra panels of electron density that further highlight the quality of the reconstruction.

*Concerning the statement, "The model also fitted unambiguously into an existing reconstruction of SAGA to reveal its binding interface and integration within the complex". The work as written hints at this interface, but doesn't really make concrete, testable predictions. Based on the crosslinking data depicted in Figure 3, which parts of Tra1 and the rest of SAGA are touching?*

We thank the reviewers for alerting us to this flaw. We have expanded our description of the Tra1-SAGA interface, by listing the precise residues within Tra1 that were crosslinked to SAGA and the subunits they pertain to, and integrating this data into Figure 3. The crosslinked residues that are closest to the observed Tra1-SAGA interface frequently target Taf12, hence we have suggested that Taf12 forms the SAGA side of this interface. We have also slightly enlarged the regions of Tra1 that are visibly in contact with SAGA, to repeats H41-H44 (residues 2150-2350) and T1-T7 (residues 2572-2830) as our original descriptions were overly specific given the resolution of the negative stain reconstruction by Durand et al. Hence we have updated the text with the proposal that Tra1 repeats T1-T7 and H41-H44 make contact to SAGA subunit Taf12, and also likely to Ada1 based on findings from the literature.

*Figure 3—figure supplement 1. It would be better to reproduce the pertinent image from Chittuluru et al. 2011 (with permission, of course), rather than just reference it. Alternatively, if a 3D EMDB model is available, it could be back-projected for comparison.*

We have received permission from Nature Publishing Group to reproduce the image from Chittuluru et al. 2011 and have included it in our revised manuscript. We could find no corresponding map deposited in the EMDB.

*Figure 4. It is surprising that the mutations, which disrupt activator targeting, are so broadly distributed throughout Tra1. The authors should further elaborate in their Discussion as to why they think this is. One would suspect that some are specific, some allosteric, and some lead to general misfolding/instability. Do any mutations cross-disrupt targeting between different activators? Can the likely effects of the mutations be categorized?*

We have elaborated on the distribution of the activator-disrupting mutations both in the Results and Discussion, suggesting that Tra1 contains multiple activator binding sites, and that these mutations abrogate these sites. We have also discussed the utility of multiple sites for transcriptional regulation. Crucial to this argument is the fact that these mutations only effect specific activators and are also spatially separated. To our knowledge, the only mutations that cross-disrupt multiple activators are those described by Knutson et al. (2011) which affect both Gcn4 and Rap1, but do not disrupt Gal4 (Lin et al. (2012)), hence we believe some activators may target overlapping/common Tra1 sites, whereas others target specific sites.

Nevertheless, our theory that these mutations abrogate distinct binding sites remains speculative, and we have acknowledged that they may instead have allosteric or stability effects, and that these mechanisms are not necessarily mutually exclusive. We favour that they mostly abrogate activator binding sites, since we posit that allosteric or stability effects would likely influence Tra1 function for all activators, rather than the selective effects that are observed for the mutations that target Gal4 vs. Gcn4/Rap1.